ecology/genetics/genomics

coral disease, gene expression, *Montastraea cavernosa*, Flower Garden Banks National Marine Sanctuary

**Author for correspondence:**
Rachel M. Wright
e-mail: rwright@smith.edu

# Gene expression associated with disease resistance and long-term growth in a reef-building coral

Emma R. Kelley[1], Robin S. Sleith[1], Mikhail V. Matz[2] and Rachel M. Wright[1,2]

[1]Department of Biological Sciences, Smith College, Northampton, MA, USA
[2]Department of Integrative Biology, University of Texas at Austin, Austin, TX, USA

RMW, 0000-0002-5867-1224

Rampant coral disease, exacerbated by climate change and other anthropogenic stressors, threatens reefs worldwide, especially in the Caribbean. Physically isolated yet genetically connected reefs such as Flower Garden Banks National Marine Sanctuary (FGBNMS) may serve as potential refugia for degraded Caribbean reefs. However, little is known about the mechanisms and trade-offs of pathogen resistance in reef-building corals. Here, we measure pathogen resistance in *Montastraea cavernosa* from FGBNMS. We identified individual colonies that demonstrated resistance or susceptibility to *Vibrio* spp. in a controlled laboratory environment. Long-term growth patterns suggest no trade-off between disease resistance and calcification. Predictive (pre-exposure) gene expression highlights subtle differences between resistant and susceptible genets, encouraging future coral disease studies to investigate associations between resistance and replicative age and immune cell populations. Predictive gene expression associated with long-term growth underscores the role of transmembrane proteins involved in cell adhesion and cell–cell interactions, contributing to the growing body of knowledge surrounding genes that influence calcification in reef-building corals. Together these results demonstrate that coral genets from isolated sanctuaries such as FGBNMS can withstand pathogen challenges and potentially aid restoration efforts in degraded reefs. Furthermore, gene expression signatures associated with resistance and long-term growth help inform strategic assessment of coral health parameters.

## 1. Introduction

Infectious diseases associated with a variety of bacterial, viral and fungal pathogens (reviewed in [1]) cause mass coral mortality

worldwide, especially in the Caribbean where stony coral tissue loss disease (SCTLD) has massively reduced live coral cover [2–4]. Several *Vibrio* species contribute to coral diseases, though the exact etiological agents for many outbreaks, including SCLTD, are uncharacterized [5,6]. Heterogeneity in disease outcomes exists between and within coral species. For example, acroporid and pocilloporid coral species appear to be among the most vulnerable taxa [7] while massive corals, like *Porites*, resist bacterial challenge [8]. These species-level differences in disease resistance shape reef communities [9,10]. Variation in disease susceptibility observed among members of a coral species (e.g. [11–13]) may contribute to reef restoration if resistant genets can repopulate degraded reefs [14].

At 190 km off the Louisiana–Texas coastline, healthy corals in Flower Garden Banks National Marine Sanctuary (FGBNMS) produce larvae that can disperse throughout the Caribbean [15]. This deep and isolated reef environment has maintained greater than 50% coral cover with no documented disease outbreaks [16]. However, a highly localized mortality event occurred in late July 2016, affecting 5.6 ha (2.6% of the area) of the East Flower Garden Bank (FGB) while the West FGB remained unaffected [17]. Diverse invertebrates presented with advancing lesions of tissue loss that mimic an infectious disease. Studies found that localized hypoxia contributed to this disease-like mortality rather than a specific bacterial pathogen [17,18]. Given the importance of this sanctuary as a source population to help restore Caribbean reefs, it is critical to assess the ability of its coral inhabitants to withstand disease challenges.

Here, we measure variation in susceptibility to a bacterial pathogen in the great star coral *Montastraea cavernosa* from FGBNMS. Once considered among the most robust Caribbean species [19], *M. cavernosa* have experienced substantial mortality from SCTLD in recent years [3]. In addition to assessing susceptibility as the appearance of tissue loss upon challenge with *Vibrio* spp., we also measure long-term calcification to account for trade-offs between growth and resistance. This coral species has demonstrated a stable calcification rate under heat stress [20], but the impacts of disease on coral growth are unknown.

This study also characterizes predictive gene expression to identify molecular markers associated with long-term calcification and resistance to bacterial pathogen invasion. Recent studies have made progress identifying allelic variation associated with coral thermal tolerance by sequencing hundreds of corals and often relying on reproductive crosses [20–23]. Here, we rely on global gene expression, which can be used to associate gene expression with complex phenotypes in wild populations [24,25]. Previous studies have identified inter-individual differences in gene expression under benign conditions that may contribute to observed differences in stress responses [26,27]. Baseline gene expression associated with resistance to bacterial challenge and calcification over the subsequent year enhance our understanding of the molecular determinants of disease resistance and growth in coral.

# 2. Methods

## 2.1. Coral collection and fragmentation

Coral fragments were obtained from the East and West Texas FGBs on 10 November 2014. SCUBA divers retrieved *M. cavernosa* fragments using hammers and chisels. Fifteen colonies were sampled from both the East and West Banks for a total of 30 colonies. The larger sampled coral fragments were then divided with a wet saw into control and experimental series ($n = 2$–3 fragments per genet per series; mean ± s.d. area = $4.8 ± 1.8$ cm$^2$). The replicate fragments recovered in 15 gallon tanks of 32 ppt artificial seawater (ASW; Instant Ocean). Tanks were maintained at 23°C under 12 000 K LED lights on a 12 L : 12 D cycle. Corals were fed Coral Frenzy every 2 days. After 2 days of recovery, fragments were moved to individual experimental chambers containing 300 ml ASW under the same lighting and temperature conditions.

## 2.2. Bacterial challenge and survival

Single isolates of *Vibrio coralliilyticus* or *Vibrio shiloi* were incubated overnight in Difco Marine Broth-2216 (BD) along with a sterile broth control at 30°C with shaking (150 rpm). Overnight cultures were triple washed in sterile ASW by centrifugation at 5000 $g$ for 10 min and resuspension in ASW. Corals were challenged with either $10^7$ CFU ml$^{-1}$ of triple-washed *Vibrio* (treatment) or the same volume of triple-washed marine broth (control) on 28 November and subsequently every 24 h for 17 days. We challenged the corals with *V. shiloi* for the first 10 days and *V. coralliilyticus* for the subsequent week

of bacterial inoculations. These bacteria were selected for their suspected coral pathogenicity [28,29]. However, some studies report that most corals remain visually healthy after experimental challenges with these pathogens [30,31]. One explanation for the inability to reliably induce symptoms of disease with isolated pathogenic agents is that virulence may depend on interactions between multiple bacterial taxa [32]. Our successive bacterial challenge design aimed to (i) provide an opportunity for bacterial interactions that may induce virulence *in situ* and (ii) increase stress incrementally in order to resolve between more susceptible and more resistant genets. Similarly, the temperature was ramped to 29°C between the fifth and sixth days of inoculation (both for controls and bacteria-inoculated fragments), as increased temperatures have been shown to induce *Vibrio* virulence [33]. The fragments were photographed daily using a Nikon D5100 with a coral health card to monitor lesion development. Time-of-death was recorded as the day when tissue loss exceeded 50% of the surface area. After the seventeenth day of bacterial challenge, surviving genets were placed in 15 gallon aquaria under control conditions.

## 2.3. Growth measurements and analysis

The number of polyps was counted on each of the surviving fragments in February 2015 and again a year later, in March 2016. Surviving individuals were weighed in February 2015 and again in February 2016 following the buoyant weight protocol [24,34]. Temperature and lighting conditions remained constant over the long-term monitoring period (25°C, 12 L : 12 D). Surface areas for each fragment were measured using IMAGEJ [35]. Skeletal weight and polyp growth were normalized to the surface area. Statistical analyses were conducted in R version 3.6.1 [36]. The R package *MCMCglmm* [37] was used to fit generalized linear mixed models for tissue and skeletal growth rates between phenotype (resistant versus susceptible), treatment (control versus *Vibrio*-challenged) and collection location (East versus West Bank).

## 2.4. Predictive gene expression

RNA was isolated from two replicate subsamples of each genet before bacterial challenge using the RNAqueous Total RNA Isolation Kit (Invitrogen). Genet identities were subsequently modified to reflect the presence of clones detected by sequence analysis, but all samples were retained in downstream gene expression analyses because each library was prepared from RNA derived from independent tissue samples. A total of 54 gene expression libraries prepared following the TagSeq protocol [38] were of high enough quality for Illumina HiSeq 2500 sequencing (SRA: PRJNA355872). Adapter sequences were trimmed and low-quality reads (minimum quality score = 20; minimum per cent bases above minimum quality score = 90%) were filtered using FASTX toolkit [39]. Reads were mapped to a holobiont reference consisting of the *M. cavernosa* genome [22] and *Cladocopium goreaui* transcriptome [40] using BOWTIE 2 [41]. Reads were converted to counts representing the number of independent observations of a transcript over all isoforms for each gene.

Isogroups (henceforth called 'genes') with a mean count of less than three across all samples were removed from the analysis. Expression sample outliers were detected using *arrayQualityMetrics* [42]. Differentially expressed genes (DEGs) were identified using *DESeq2* [43]. Wald tests were performed to compare phenotype (resistant versus susceptible) and collection location (EastFGB versus WestFGB) using the model 'count~susceptibility phenotype + bank' ($n = 50$ after outlier detection). Wald tests were also performed to compare continuous growth phenotypes using the models 'count~mean calcification rate + mean polyp generation + susceptibility phenotype' ($n = 38$ after outlier detection). The DESeq2 models were run independently as we necessarily had slightly different samples in each model owing to limitations in the ability to collect growth data from susceptible corals that did not survive the year-long growth monitoring period. We report Wald statistics (log fold change/standard error) to represent the magnitude of expression difference between groups or per unit change of continuous variables. False-discovery rate (FDR) *p*-values were adjusted using the Benjamini–Hochberg procedure [44]. Gene expression heatmaps were generated using *pheatmap* [45] and gene ontology enrichment was performed based on signed adjusted *p*-values using *GO-MWU* [46].

## 2.5. Reference-based 2bRAD genotyping

We prepared 64 genotyping libraries using the 2bRAD protocol [47] and sequenced the libraries on the Illumina HiSeq 2000 platform at UT Austin Genome Sequencing and Analysis Facility. We used FASTX

toolkit to remove barcodes, deduplicate reads and apply quality filters such that only reads in which 90% or more of the bases with a Phread score $\geq 20$ were retained. These reads were mapped to the *M. cavernosa* genome [22] using Bowtie 2 [41]. Genotyping was performed with ANGSD v0.930 [48]. Sites were filtered to retain loci with a mapping quality $\geq 20$, base call quality $\geq 30$ and minor allele frequency $\geq 0.05$ that were sequenced in at least 20 individuals. These sites were used to calculate pairwise identity-by-state distances between individual samples. Distances $< 0.15$ were presumed to be clones based on similarity detected across genotyping replicates. Only one clone per sample was retained for subsequent population genetic analysis. Library replicates were removed as clones, as well as two additional pairs of clones. The VCFtools subprogram *weir-fst-pop* calculated fixation index ($F_{ST}$) estimates. To determine dominant symbiont types, we mapped 2bRAD sequences to a combined symbiont reference composed of transcriptomes from *Symbiodinium* 'clades' A and B [49] and 'clades' C and D [50] using a custom perl script `zooxtype.pl`. Custom scripts are hosted within the 2bRAD GitHub repository (https://github.com/z0on/2bRAD_denovo).

# 3. Results

## 3.1. 2bRAD genotyping

An average of 71.9% of reads uniquely mapped to the *M. cavernosa* genome across the 64 2bRAD libraries and an average of 67.3% of sites were covered at greater than 5× sequencing depth. We identified two pairs of clones (7/29 and 17/21; electronic supplementary material, figure S1), which are presumably fragments inadvertently sampled from different parts of the same colony. We identified 11 081 single nucleotide polymorphisms (SNPs) from 26 unique (non-clonal) samples. Principal component analysis based on identity-by-state demonstrates a lack of genetic structure between sampling sites or resistance phenotype groups (electronic supplementary material, figure S2). $F_{ST}$ represents the level of genetic differentiation between groups. We calculated weighted $F_{ST}$, which accounts for differences in the numbers of individuals in each group. We observed no genetic differentiation between corals grouped by resistance phenotype ($F_{ST} = 0$) and little genetic differentiation between East and West FGB origin ($F_{ST} = 0.004$). Mapping 2bRAD-seq data to symbiont references determined that all corals were dominated by *Cladocopium* (electronic supplementary material, figure S3).

## 3.2. Bacterial challenge survival

Time-of-death was recorded when a coral fragment displayed greater than 50% tissue loss (e.g. figure 1a). Survival ranged from 10 to 22 days (mean ± s.d. 18.6 ± 3.3 days) among fragments that developed lesions (figure 2). Only one fragment from genet 10 developed a lesion and subsequently recovered (electronic supplementary material, figure S4). Bacterial challenge significantly increased mortality ($p = 2 \times 10^{-16}$; figure 1b). Collection site was not associated with differential survival ($p = 0.13$; figure 1c).

## 3.3. Long-term coral growth

Across all coral fragments, the average (±s.d.) long-term calcification rate was $135 \pm 61$ mg cm$^{-2}$ yr$^{-1}$. We observed between 0 and 30 new polyps on each fragment over the year (mean ± s.d. = 10.9 ± 6.7), which ranged from approximately 0–7 new polyps cm$^{-2}$ of surface area. Neither annual calcification rate nor annual polyp generation was significantly associated with collection site, treatment or resistance phenotype (figure 3).

## 3.4. Host differential gene expression

TagSeq yielded an average 319 547 *M. cavernosa* (coral host) gene counts per sample after filtering lowly expressed genes (base mean < 3). No genes were differentially expressed according to sampling origin and only one unannotated gene was significantly associated with an annual increase in polyp number (Mcavernosa00313, Wald stat = −4.8, FDR = 0.011).

The contrast between resistant and susceptible genets yielded only one DEG at FDR = 0.05. This transcript shares substantial homology with a lymphocyte antigen 6H-like gene identified in *Orbicella faveolata* (E-value = $2 \times 10^{-141}$, identity = 87%) and was more highly expressed in resistant genets (Wald stat = 6.2, FDR = $5.7 \times 10^{-6}$). Gene ontology (GO) enrichment tests between resistant and susceptible

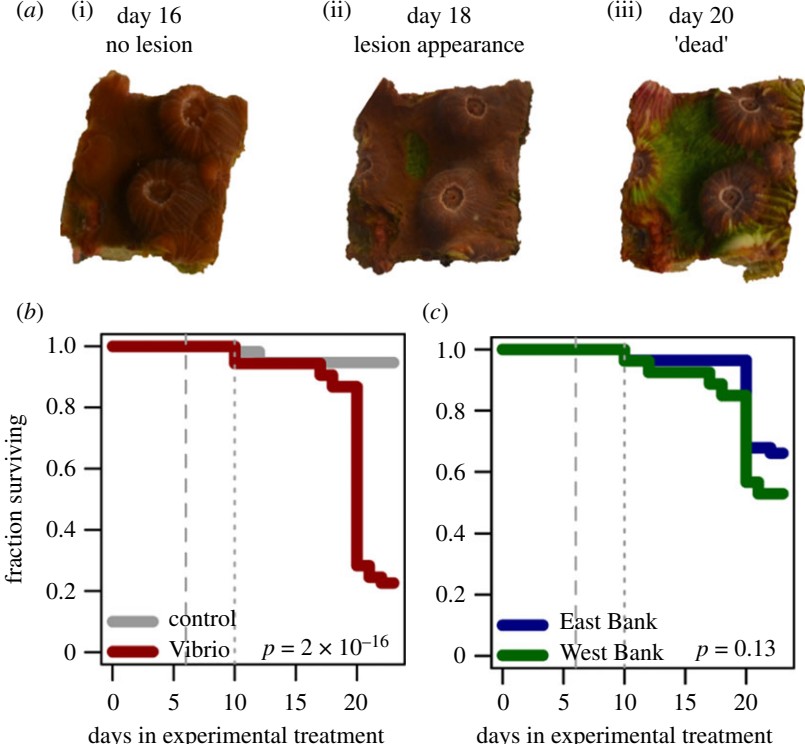

**Figure 1.** (*a*) Example lesion progression. (*b*) The survival of coral fragments in the control (grey) or *Vibrio* treatments (red). (*c*) The survival of coral fragments from East FGB (blue) or West FGB (green). *p*-values correspond to the effect of treatment (*b*) or collection site (*c*) in a Cox proportional hazards model. The dashed line at day 6 denotes a shift from 23℃ to 29℃. The dotted line at day 10 and a shift from *Vibrio shiloi* to *Vibrio coralliilyticus* exposure.

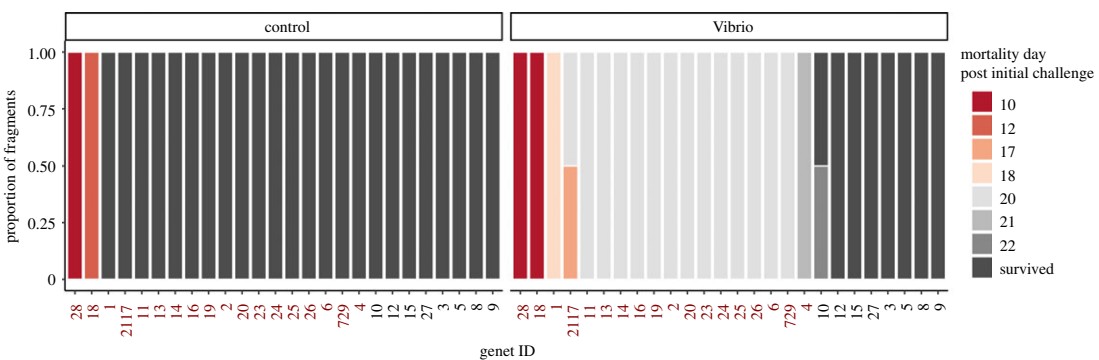

**Figure 2.** Timing of mortality for genets in the control and *Vibrio* treatments. Susceptible and resistant genet identities are shown in red and black, respectively. Columns are ordered left to right from earliest to latest mortality.

corals yielded eight biological processes (BP), 20 cellular components (CC) and five molecular functions (MF) significantly enriched (adjusted $p < 0.05$) among genes associated with disease phenotype. Among these terms, several categories related to cell division (e.g. regulation of mitotic cell cycle, DNA integrity checkpoint, kinetochore microtubule) were enriched with genes showing higher expression in resistant corals (figure 4).

Fifty-nine DEGs were significantly associated with the rate of calcification in the year following TagSeq library preparation (electronic supplementary material, Data S1). Some of the top DEGs associated with calcification rate include those encoding Hairy and Enhancer of Split (HES) 1 (Mcavernosa12226, Wald stat = 3.8, FDR = 0.032, *E*-value = 0.0) and putative cell adhesion proteins fermitin family homologue 2 (Mcavernosa09908, Wald stat = 5.1, FDR = $5.1 \times 10^{-4}$, *E*-value = $1 \times 10^{-24}$) and coadhesin (Mcavernosa11572, Wald stat = 4.3, FDR = $7.4 \times 10^{-3}$, *E*-value = 0.0) (figure 5). GO enrichment tests yielded 12 BP, 14 CC and 17 MF significantly enriched (adjusted $p < 0.05$) with genes associated with calcification rate (electronic supplementary material, figure S5A). Among these terms,

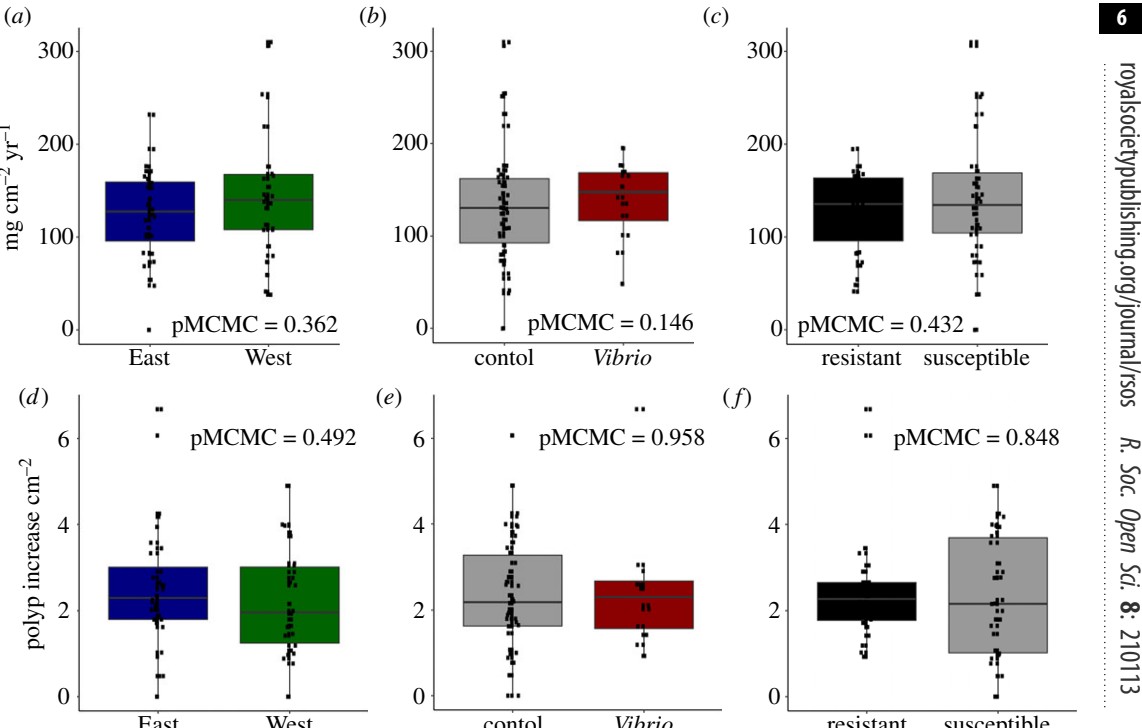

**Figure 3.** Calcification rate (mg cm$^{-2}$ yr$^{-1}$; top) was not associated with collection site (*a*), treatment (*b*) or resistance phenotype (*c*). Polyp generation (new polyps cm$^{-2}$; bottom) was not also associated with collection site (*d*), treatment (*e*) or resistance phenotype (*f*).

'intrinsic component of the plasma membrane' and 'regulation of organ morphogenesis' were enriched with genes showing higher expression in corals with faster annual calcification rates.

Seventeen DEGs were significantly associated with the rate of new polyp generation in the year following TagSeq library preparation (electronic supplementary material, Data S1). Some of the top DEGs associated with polyp generation include transcripts encoding a heat shock protein chaperone sacsin (Mcavernosa09333, Wald stat = 4.0, FDR = 0.03, *E*-value = 0.0) and major facilitator superfamily domain-containing protein 12 (Mcavernosa16593, Wald stat = −4.3, FDR = 0.018, *E*-value = $5 \times 10^{-29}$). The fermitin family homologue 2 transcript that was associated with increased calcification rate was significantly downregulated in corals that generated more new polyps (Wald stat = −4.8, FDR = $3.6 \times 10^{-3}$) (figure 5; electronic supplementary material, Data S1). GO enrichment tests yielded 20 BP, 12 CC and 10 MF significantly enriched (adjusted $p < 0.05$) with genes associated with polyp generation (electronic supplementary material, figure S5B). Among these terms, 'nuclear speck' and 'mRNA processing' were enriched with genes showing higher expression in corals that generated more polyps over the subsequent year.

## 3.5. Algal symbiont gene expression

TagSeq yielded an average of 9510 *Cladocopium* (algal symbiont) counts per sample after filtering lowly expressed genes (base mean < 3). No symbiont genes were significantly associated with sampling location, host calcification rate or host polyp generation (FDR = 0.05). One unannotated gene was significantly associated with host resistance to bacterial challenge (Wald stat = 4.6, FDR = 0.003). GO enrichment analyses did not yield any significantly enriched categories for the symbiont genes.

# 4. Discussion

## 4.1. No trade-offs between resistance and coral growth

Variation in disease resistance can be explained by differential investment in immunity parameters [19] that compete for energetic resources with other life-history traits such as growth and reproduction [51].

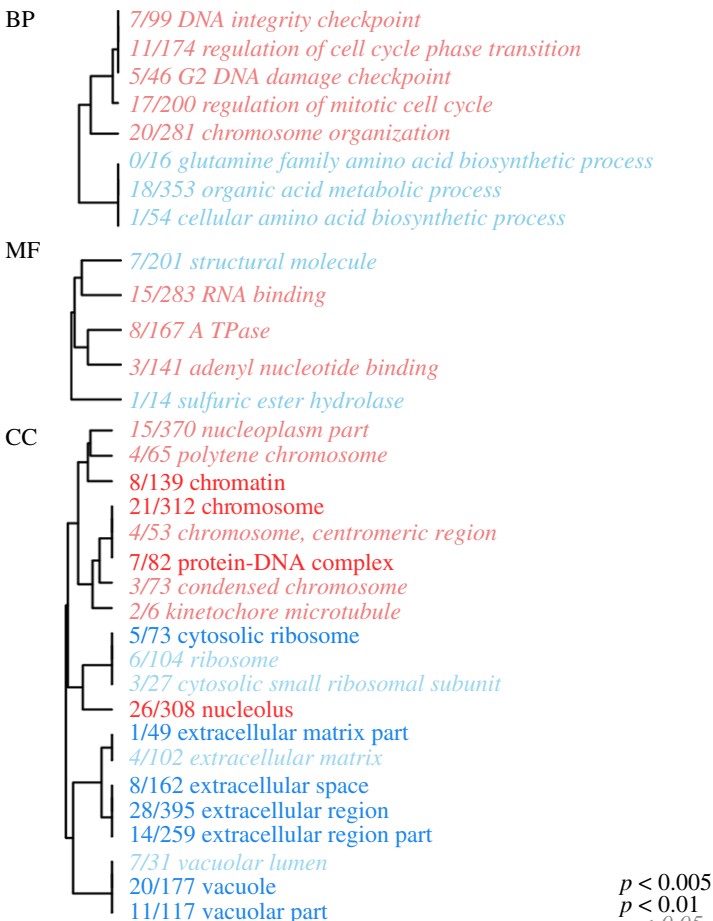

**Figure 4.** BP, MF and CC enriched by adjusted *p*-value generated by testing for association with resistance phenotype. The text colour indicates the direction of differential expression between resistant and susceptible genets (red, upregulated in resistant corals; blue, upregulated in susceptible corals). The text size indicates the significance of the term as indicated by the inset key. The fraction preceding the term indicates the number of genes within the term that had an adjusted *p*-value less than 0.05. Trees indicate gene sharing among GO categories (categories with no branch length between them are subsets of each other).

Here, we found no association between long-term growth (polyp generation or buoyant weight increase) and disease resistance (figure 3*c,f*). This result complements previous findings that growth rates in another reef-builder, *Acropora millepora*, were not associated with trade-offs in other health parameters, including survival under *Vibrio* challenge [12]. We also found that surviving corals demonstrated similar long-term growth rates regardless of whether they received sterile media or *Vibrio* culture during the experimental period (figures 3 and 5*b,e*). These results suggest that long-term growth rates can remain stable after a disease event if a coral can survive and recover from an outbreak. However, back-to-back bleaching events [52] and multi-year infectious disease outbreaks [3] limit the amount of time a coral can recover before the next life-threatening challenge. Furthermore, our long-term growth data summarize growth over an entire year, so we cannot detect any sharp variation in growth rate that may have occurred immediately following the disease challenge. Future studies could investigate the immediate effects of bacterial challenge on coral growth with more frequent measurements.

Our experimental design included bacterial challenges with two *Vibrio* species and a temperature increase in an effort to create an increasingly stressful environment that could reveal differences in susceptibility among the coral genets. However, our results demonstrate that our coral genets were only susceptible to the successive challenge at increased temperature. A recent study [53] was able to induce mortality in fragments of apparently healthy *M. cavernosa* from the disease-affected Florida Reef Tract within a 21-day time period, which is consistent with the timing of lesion onset in this study (10–22 days after initial infection; mean ± s.d. = 18.6 ± 3.3 days). Future studies may consider including multiple populations of corals with suspected variance in disease susceptibility to better discern patterns of expression related to bacterial resistance. These studies may also take care to

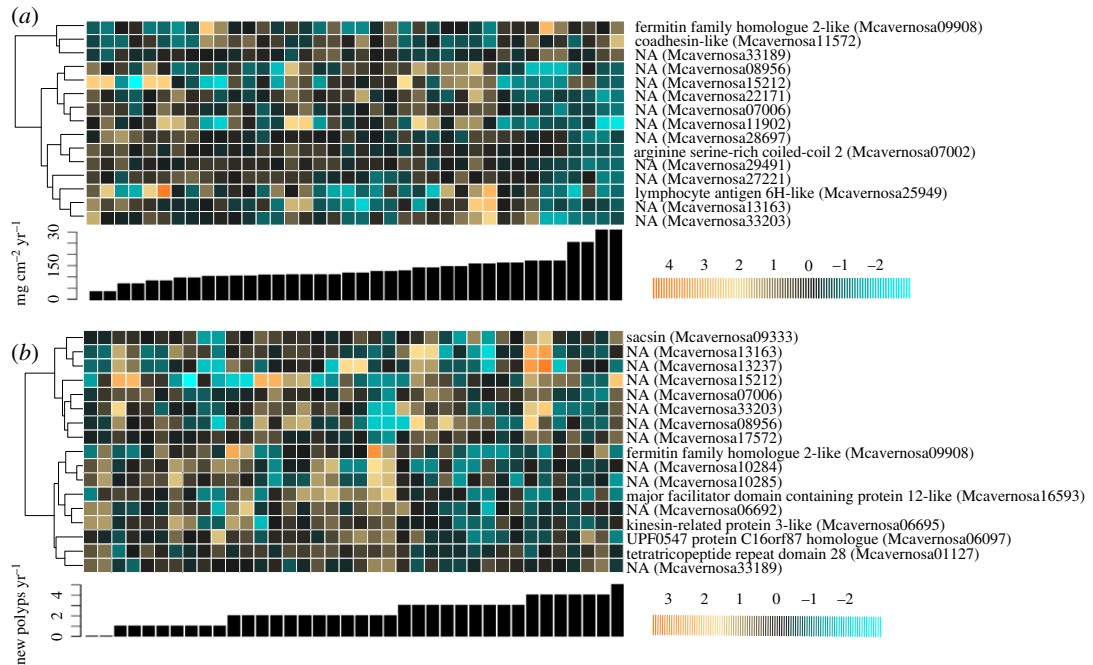

**Figure 5.** Gene expression associated with (*a*) annual calcification rate (FDR < 0.01) and (*b*) annual polyp generation (FDR < 0.05). Heatmap rows are genes, and columns are samples. Samples are ordered by calcification rate, as indicated by the bar heights. The colour scale is in log2-fold change relative to the gene's mean. Genes are hierarchically clustered based on Pearson's correlations across samples.

monitor lesion progression rates, in addition to lesion onset and 'time-of-death', in order to provide more precise metrics of disease risk.

## 4.2. Genomic associations with disease resistance

One objective of the TagSeq experiment was to associate predictive gene expression with subsequent exposure outcome. We did not attempt to find differences in gene expression responses between individuals, and, in this case, we would be unable to do so given the level of mortality we observed. The remaining tissue from the susceptible individuals at 50% lesion progression was also beginning to slough. Still, identifying differences in gene expression responses between susceptible and resistant individuals is critical to understanding molecular mechanisms underlying disease tolerance. Future studies attempting to fill that knowledge gap will need to carefully consider when to sample for expression. For example, recent work investigating responses to thermal stress in corals and symbiotic anemones has shown that symbiosis breaks down well before bleaching becomes apparent [54,55]. Furthermore, future studies should note that our ability to detect predictive gene expression associated with a subsequent bacterial challenge outcome was limited by our sample size ($n = 26$ genets in the gene expression analysis). Collection limits for threatened species and experimental constraints, such as isolated aquaria needed to prevent coral-to-coral disease transmission, restrict the number of genets that can be included in similar experiments. We recommend careful consideration of the sample size necessary to detect the signature of interest given the large variation in constitutive gene expression among individuals.

Our study to identify predictive gene expression associations with disease outcome revealed subtle differences in pre-exposure transcriptomic states between corals that subsequently demonstrated resistance or susceptibility to *Vibrio* challenge. Only one transcript, which shares extensive homology with a lymphocyte antigen-6 (Ly6) gene, passed the genome-wide significance threshold of FDR = 0.05; it was upregulated in resistant corals prior to bacterial challenge (electronic supplementary material, Data S1). Genes belonging to the Ly6 family play various roles across metazoans, such as epithelial barrier formation in *Drosophila* [56] and neutrophil migration in mammals [57]. In mouse epithelial cells, the expression of a Ly6 protein (Lypd8) promotes gut homeostasis and prevents pathogen attachment [58]. Given that many coral diseases are associated with the loss of tissue

structure and bacterial infiltration [59], future studies should explore the potential role of this gene family in promoting coral tissue integrity upon bacterial challenge.

Mitotic activity including spindle formation and cell cycle phase regulation were enriched among upregulated genes in resistant individuals (figure 4), possibly indicating an abundance of a specific population of proliferative cells or higher cell division rates in resistant individuals, though our bulk transcriptomic data cannot discern between these hypotheses. Future investigations using single-cell transcriptomics may reveal differences in activated cell populations between resistant and susceptible genets. Alternatively, senescence may explain differences in cell growth rates [60]. Colony age can explain differences in disease susceptibility, as has been reported in *Acropora palmata* affected with white-pox disease [61]. All coral cells of a clonal genet have the same age since sexual recruitment (i.e. chronological age), but soft tissues across the colony have probably experienced different numbers of cell divisions (i.e. replicative age). Across colonies of the massive reef-builder *Porites* with an average age of 41 years, the average polyp age was only 2–3 years [62]. Future investigations of the impacts of ageing and age-related cell turnover rates [63] on disease susceptibility in corals could evaluate markers of replicative age, such as telomere length or somatic mutation accumulation [64].

The small sample size of this study precludes investigation of the genomic architecture underlying disease resistance, though *in situ* disease transmission experiments provide evidence for a genetic basis to disease resistance in some species of reef-building corals [13,65]. A recent study that identified dozens of genetic variants associated with resistance to *Vibrio* infection in a flatfish used phenotype data from thousands of fishes and whole-genome resequencing for over 500 individuals [66]. Conducting experiments at this scale in threatened coral species presents considerable challenges, though genomic predictors for thermal tolerance in corals have been possible through low-coverage sequencing from minimally invasive tissue samples from hundreds of adult colonies *in situ* [22] and genome-wide SNP analysis of coral larvae produced through sexual reproduction from experimentally selected parent colonies [23]. Our estimates of $F_{ST}$ between corals from the East and West FGB match previous studies [67] and support models of high gene flow through larval dispersal in the region [15,68].

## 4.3. Predictive gene expression associated with long-term calcification

Transcripts homologous to HES, coadhesin and transmembrane protease serine 9 protein were more highly expressed in corals with higher annual calcification rates (electronic supplementary material, Data S1). HES regulates bone mass in mammals [69], but its role in coral biology is currently unclear. Coadhesin and transmembrane protease serine 9 have been identified as part of the coral skeletal proteome in *Acropora millepora* [70] and *Stylophora pistillata* [71], respectively. Given their demonstrated role in coral calcification in previous studies and our predictive gene expression associations here, these genes represent prime candidates for validation as potential predictive growth biomarkers.

# 5. Conclusion

We demonstrate intraspecific variation in pathogen resistance in a reef-building coral from an isolated marine sanctuary with no documented instance of coral disease. Understanding the immediate and long-term consequences of bacterial pathogen exposure is especially important given the potential impact of this sanctuary as a larval source to restore disease-degraded Caribbean coral populations. The presence of resistant genets and lack of trade-offs between resistance and growth under these laboratory conditions provide hope that this coral population may be able to withstand some bacterial challenge. However, ever-worsening ocean conditions threaten marine organisms with multiple concurrent stressors. The health of coral reefs ultimately relies on global action to mitigate the effects of climate change.

Data accessibility. Sequences are deposited in the SRA PRJNA355872 (TagSeq) and PRJNA655196 (2bRAD). Phenotype data and analysis scripts are hosted at https://github.com/rachelwright8/McavSusceptibility and have been archived within the Zenodo repository: http://doi.org/10.5281/zenodo.4633245. Custom 2bRAD scripts are hosted within the 2bRAD GitHub repository (https://github.com/z0on/2bRAD_denovo).
Authors' contributions. E.R.K. analysed the phenotypic and gene expression data and contributed to manuscript preparation. R.S.S. participated in data analysis and contributed to manuscript preparation. M.V.M. coordinated the study and critically revised the manuscript. R.M.W. conceived of and designed the study, collected coral specimens, carried out the bacterial challenge and molecular work, analysed the 2bRAD sequence data, and drafted the manuscript.

Competing interests. We declare we have no competing interests.

Funding. Funding was provided by a Texas State Aquarium grant awarded to M.V.M. and support from the Nancy Kershaw Tomlinson Fund for Undergraduate Honors Research to E.R.K.

Acknowledgements. The authors are grateful to the NOAA research team at FGBNMS, including Emma Hickerson and the crew of the *R/V Manta* for field assistance. These corals were collected under permit FGBNMS-2014-0013. We thank the Texas Advanced Computing Center for computational resources and to Dr Laura Katz for mentorship.

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
