## [Peer Review File · Royal Society Open Science]

Review History

RSOS-210113.R0 (Original submission)

Review form: Reviewer 1

Is the manuscript scientifically sound in its present form?

Yes

Are the interpretations and conclusions justified by the results?

Yes

Is the language acceptable?

Yes

Do you have any ethical concerns with this paper?

No

Have you any concerns about statistical analyses in this paper?

No

Recommendation?

Accept as is

Comments to the Author(s)

The authors have clearly taken the time to provide thoughtful and thorough responses to my previous comments. I thank them for the effort. I am truly excited to see this manuscript published.

Review form: Reviewer 2 (Francois Seneca)**Is the manuscript scientifically sound in its present form?**

Yes

Are the interpretations and conclusions justified by the results?

No

Is the language acceptable?

Yes

Do you have any ethical concerns with this paper?

No

Have you any concerns about statistical analyses in this paper?

Yes

Recommendation?

Major revision is needed (please make suggestions in comments)

Comments to the Author(s)

Dear authors,

this manuscript is well written and of appropriate length. You set this paper to address the question of mechanisms and trade-offs of pathogen resistance in corals, which is a very interesting and important question for the way we will engage in protecting/preserving coral reefs in the future. However, it is my opinion that you unfortunately fall short of producing results that would truly help advancing this question. First of all, you identified genets that were resistant but did not really measure "pathogen resistance" in these individuals. Eight out of 26 genets were capable of resisting the bacterial and heating assaults, but if I am correct gene expression levels were not assessed during or after the bacterial exposures. It seems that you expected differences in constitutive gene expression levels during normal conditions among your genets? Why was that? Did you base this hypothesis on certain prior observations? Size, color, location, environmental conditions, fitness, anything? And then, if you were looking for correlations between gene expression levels and resistance/susceptibility to bacterial infection, why did you not collect tissue during the experiment, for example after the 50% tissue loss had been reached, or from replicated fragments made before the start of the experiment? This is a real missed opportunity to produce valuable gene expression data that would allow us to discover what genes may be involved in making certain genets resistant. I highly encourage you to pursue this question and try again by setting up an experiment that will get you some results. Here, one differentially expressed gene between resistant and susceptible genets cannot be yielding significant GO enrichment! I do not understand how you get to those results? The result that

resistant genets do not seem to show a decrease in growth rate after bacterial exposure is interesting, but it would have been nice to clearly see those results in the paper. Is there really no growth difference between the control and exposed resistant fragments after a year? Finally, some of your main conclusions are unfortunately not supported by your results. When you state : "Predictive (pre-exposure) gene expression highlights subtle differences between resistant and susceptible genets" and "Furthermore, gene expression signatures associated with resistance and long-term growth help inform strategic assessment of coral health parameters.", it is all based on that one single differentially expressed gene. This lack of gene detection is not because of a technical error, but because you are most likely not looking in what/when you need to. Because the pathogen resistance side of the paper is half of the story, I cannot recommend your manuscript for publication in the present form. I know it is disappointing, but I would suggest completing this work with data that compare gene expression levels between resistant and susceptible genets under bacterial assault, or re-write the paper to bring the predictive genes and long-term growth rate correlation side in the fore front. But, I would agree it does not constitute a very compelling story in itself without additional work.

Best wishes

Decision letter (RSOS-210113.R0)

Dear Ms Wright

The Editors assigned to your paper RSOS-210113 "Gene expression associated with disease resistance and long-term growth in a reef-building coral" have now received comments from reviewers and would like you to revise the paper in accordance with the reviewer comments and any comments from the Editors. Please note this decision does not guarantee eventual acceptance.

One reviewer recommends immediate acceptance, while the other reviewer raises a number of issues which will require careful consideration. We invite you to respond to the comments supplied below and revise your manuscript. Below the referees' and Editors' comments (where applicable) we provide additional requirements. Final acceptance of your manuscript is dependent on these requirements being met. We provide guidance below to help you prepare your revision.

Please submit your revised manuscript and required files (see below) no later than 21 days from today's (ie 23-Feb-2021) date. Note: the ScholarOne system will 'lock' if submission of the revision is attempted 21 or more days after the deadline. If you do not think you will be able to meet this deadline please contact the editorial office immediately.

on behalf of Prof Steve Brown (Subject Editor)
openscience@royalsociety.org

Associate Editor Comments to Author:

Thank you for so positively engaging with the commentary from the review at our sister journal PRSB. The reviewer from that journal recommends acceptance, though a second (and new) reviewer for RSOS has a number of queries that we'd like you to address - we will invite that reviewer to take a look at your revision. We'll look forward to receiving the revision in the near future.

Reviewer comments to Author:

Reviewer: 1

Comments to the Author(s)

The authors have clearly taken the time to provide thoughtful and thorough responses to my previous comments. I thank them for the effort. I am truly excited to see this manuscript published.

Reviewer: 2

Comments to the Author(s)

Dear authors,

this manuscript is well written and of appropriate length. You set this paper to address the question of mechanisms and trade-offs of pathogen resistance in corals, which is a very interesting and important question for the way we will engage in protecting/preserving coral reefs in the future. However, it is my opinion that you unfortunately fall short of producing results that would truly help advancing this question. First of all, you identified genets that were resistant but did not really measure "pathogen resistance" in these individuals. Eight out of 26 genets were capable of resisting the bacterial and heating assaults, but if I am correct gene expression levels were not assessed during or after the bacterial exposures. It seems that you expected differences in constitutive gene expression levels during normal conditions among your genets? Why was that? Did you base this hypothesis on certain prior observations? Size, color, location, environmental conditions, fitness, anything? And then, if you were looking for correlations between gene expression levels and resistance/susceptibility to bacterial infection, why did you not collect tissue during the experiment, for example after the 50% tissue loss had

been reached, or from replicated fragments made before the start of the experiment? This is a real missed opportunity to produce valuable gene expression data that would allow us to discover what genes may be involved in making certain genets resistant. I highly encourage you to pursue this question and try again by setting up an experiment that will get you some results. Here, one differentially expressed gene between resistant and susceptible genets cannot be yielding significant GO enrichment! I do not understand how you get to those results? The result that resistant genets do not seem to show a decrease in growth rate after bacterial exposure is interesting, but it would have been nice to clearly see those results in the paper. Is there really no growth difference between the control and exposed resistant fragments after a year? Finally, some of your main conclusions are unfortunately not supported by your results. When you state: "Predictive (pre-exposure) gene expression highlights subtle differences between resistant and susceptible genets" and "Furthermore, gene expression signatures associated with resistance and long-term growth help inform strategic assessment of coral health parameters.", it is all based on that one single differentially expressed gene. This lack of gene detection is not because of a technical error, but because you are most likely not looking in what/when you need to. Because the pathogen resistance side of the paper is half of the story, I cannot recommend your manuscript for publication in the present form. I know it is disappointing, but I would suggest completing this work with data that compare gene expression levels between resistant and susceptible genets under bacterial assault, or re-write the paper to bring the predictive genes and long-term growth rate correlation side in the fore front. But, I would agree it does not constitute a very compelling story in itself without additional work.

Best wishes

===PREPARING YOUR MANUSCRIPT===

If you have been asked to revise the written English in your submission as a condition of publication, you must do so, and you are expected to provide evidence that you have received language editing support. The journal would prefer that you use a professional language editing service and provide a certificate of editing, but a signed letter from a colleague who is a native speaker of English is acceptable. Note the journal has arranged a number of discounts for authors

using professional language editing services
(<https://royalsociety.org/journals/authors/benefits/language-editing/>).

===PREPARING YOUR REVISION IN SCHOLARONE===

-- If you have uploaded ESM files, please ensure you follow the guidance at <https://royalsociety.org/journals/authors/author-guidelines/#supplementary-material> to include a suitable title and informative caption. An example of appropriate titling and captioning may be found at https://figshare.com/articles/Table_S2_from_Is_there_a_trade-

off_between_peak_performance_and_performance_breadth_across_temperatures_for_aerobic_sc
ope_in_teleost_fishes_/3843624.

Author's Response to Decision Letter for (RSOS-210113.R0)

See Appendix A.

RSOS-210113.R1 (Revision)

Review form: Reviewer 2 (Francois Seneca)

Is the manuscript scientifically sound in its present form?

Yes

Are the interpretations and conclusions justified by the results?

Yes

Is the language acceptable?

Yes

Do you have any ethical concerns with this paper?

No

Have you any concerns about statistical analyses in this paper?

No

Recommendation?

Accept as is

Comments to the Author(s)

Thank you for your efforts in responding to my questions and concerns. I can concede that you set this study to look for constitutive gene expression that would predict the outcome of pathogen assaults. However, as you very pertinently pointed it out, gene expression variation between corals of the same population and species is often great, therefore in order to answer your main question of interest, the sample size would have to be much larger for a chance to detect a signal (if it exists). I was just surprised this was your experimental design choice to go after that kind of question. Your paper is well written and the analyses are sound. I just remain unconvinced that your sample size was ever going to let you detect constitutive gene expression differences predictive of disease resistance among genets living in the same general environmental conditions. To give you an example, recent inflammasome markers on different immune cell populations can predict the clinical outcome of COVID patients, but these markers were evaluated on 66 people. But I also understand that balancing budget, sample size and question of interest can be tricky. Best wishes.

Decision letter (RSOS-210113.R1)

Dear Ms Wright

On behalf of the Editors, we are pleased to inform you that your Manuscript RSOS-210113.R1 "Gene expression associated with disease resistance and long-term growth in a reef-building coral" has been accepted for publication in Royal Society Open Science subject to minor revision in accordance with the referees' reports. Please find the referees' comments along with any feedback from the Editors below my signature.

The reviewer and associate editor are very positive about publication of the paper, but would like you to add a few comments in relation to the potential limitations of the work, as pointed out in the reviewer comments. We invite you to respond to the comments and revise your manuscript. Below the referees' and Editors' comments (where applicable) we provide additional requirements. Final acceptance of your manuscript is dependent on these requirements being met. We provide guidance below to help you prepare your revision.

Please submit your revised manuscript and required files (see below) no later than 7 days from today's (ie 17-Mar-2021) date. Note: the ScholarOne system will 'lock' if submission of the revision is attempted 7 or more days after the deadline. If you do not think you will be able to meet this deadline please contact the editorial office immediately.

on behalf of Proft Steve Brown (Subject Editor)
openscience@royalsociety.org

Associate Editor Comments to Author:

The editors recommend acceptance, following the comments of the reviewer, but we would also like you to respond to the remaining comments from that reviewer to more fully address the potential limitations of the work (and how these could/should be overcome in future works). This need not be an extensive re-write, but adding a few sentences to take these points into consideration would be welcome.

Reviewer comments to Author:
Reviewer: 2

Comments to the Author(s)

Thank you for your efforts in responding to my questions and concerns. I can concede that you set this study to look for constitutive gene expression that would predict the outcome of pathogen assaults. However, as you very pertinently pointed it out, gene expression variation between corals of the same population and species is often great, therefore in order to answer your main question of interest, the sample size would have to be much larger for a chance to detect a signal (if it exists). I was just surprised this was your experimental design choice to go after that kind of question. Your paper is well written and the analyses are sound. I just remain unconvinced that your sample size was ever going to let you detect constitutive gene expression differences predictive of disease resistance among genets living in the same general environmental conditions. To give you an example, recent inflammasome markers on different immune cell populations can predict the clinical outcome of COVID patients, but these markers were evaluated on 66 people. But I also understand that balancing budget, sample size and question of interest can be tricky. Best wishes.

===PREPARING YOUR MANUSCRIPT===

Your revised paper should include the changes requested by the referees and Editors of your manuscript. You should provide two versions of this manuscript and both versions must be provided in an editable format:
one version identifying all the changes that have been made (for instance, in coloured highlight, in bold text, or tracked changes);
a 'clean' version of the new manuscript that incorporates the changes made, but does not highlight them. This version will be used for typesetting.
Please ensure that any equations included in the paper are editable text and not embedded images.

===PREPARING YOUR REVISION IN SCHOLARONE===

To revise your manuscript, log into <https://mc.manuscriptcentral.com/rsos> and enter your Author Centre - this may be accessed by clicking on "Author" in the dark toolbar at the top of the

page (just below the journal name). You will find your manuscript listed under "Manuscripts with Decisions". Under "Actions", click on "Create a Revision".

<https://royalsociety.org/journals/authors/author-guidelines/#supplementary-material> to include a suitable title and informative caption. An example of appropriate titling and captioning may be found at https://figshare.com/articles/Table_S2_from_Is_there_a_trade-off_between_peak_performance_and_performance_breadth_across_temperatures_for_aerobic_sc_ope_in_teleost_fishes_/3843624.

Author's Response to Decision Letter for (RSOS-210113.R1)

See Appendix B.

Decision letter (RSOS-210113.R2)

Dear Ms Wright,

I am pleased to inform you that your manuscript entitled "Gene expression associated with disease resistance and long-term growth in a reef-building coral" is now accepted for publication in Royal Society Open Science.

on behalf of Professor Steve Brown (Subject Editor)
openscience@royalsociety.org

Appendix A

Associate Editor Comments to Author:

Thank you for so positively engaging with the commentary from the review at our sister journal PRSB. The reviewer from that journal recommends acceptance, though a second (and new) reviewer for RSOS has a number of queries that we'd like you to address - we will invite that reviewer to take a look at your revision. We'll look forward to receiving the revision in the near future.

We thank the editorial team for their continued coordination of this effort. We ask that our work be considered in the context of our stated experimental objective: to identify individual-specific gene expression signatures that can be measured prior to any treatment but would predict the coral's ability to tolerate infection better or grow faster than others. Such signatures are particularly useful as biomarkers. We are keenly aware that most gene expression studies measure a transcriptional response to an experimental challenge, but this is not the type of experiment we meant to conduct. Interindividual differences in transcriptional activity under non-stressful conditions that nevertheless contribute to stress resilience are understudied. This work seeks to fill this knowledge gap in an ecologically important species that is currently experiencing extreme population declines due to infectious disease. In this case we did not find many strong associations between gene expression and disease onset. Strong predictive genes would certainly be a more thrilling scientific story but in this case the real result, that only subtle differences in expression profile underlie differences in infection outcome, is of immediate importance to a scientific community tasked with understanding how gene expression studies may inform us about coral health risks.

Reviewer comments to Author:

Reviewer: 1

Comments to the Author(s)

The authors have clearly taken the time to provide thoughtful and thorough responses to my previous comments. I thank them for the effort. I am truly excited to see this manuscript published.

We appreciate the reviewer's careful consideration of our work and credit them for their creative and thoughtful suggestions for improving the manuscript.

Reviewer: 2

Comments to the Author(s)

Dear authors,

this manuscript is well written and of appropriate length. You set this paper to address the question of mechanisms and trade-offs of pathogen resistance in corals, which is a very interesting and important question for the way we will engage in protecting/preserving coral reefs in the future. However, it is my opinion that you unfortunately fall short of producing results that would truly help advancing this question. First of all, you identified genes that were resistant but did not really measure "pathogen resistance" in these individuals. Eight out of 26

genets were capable of resisting the bacterial and heating assaults, but if I am correct gene expression levels were not assessed during or after the bacterial exposures.

The reviewer is correct that we identified resistant and susceptible genets by monitoring the onset of lesions after experimental exposure, as has been similarly performed in many coral disease experiments (e.g., Volmer & Kline 2008, Muller, Bartels, & Baum 2018, Rosales et al., 2019, and our own previously published work, Wright et al., 2017 and Wright et al., 2019). The reviewer is also correct that we did not measure gene expression during or after bacterial exposure because our experimental objective is to associate baseline gene expression with subsequent outcomes to a bacterial exposure. Comparing post-exposure gene expression is not a stated goal of this experiment and in this case an attempt to measure gene expression responses would be counterproductive for reasons we explain in detail below.

It seems that you expected differences in constitutive gene expression levels during normal conditions among your genets ? Why was that ? Did you base this hypothesis on certain prior observations ? Size, color, location, environmental conditions, fitness, anything ?

We expected to see constitutive differences in gene expression levels during normal conditions among our genets because they are genetically distinct individuals. Gene expression profiles vary strongly among coral individuals due to genetics (Dixon et al., 2015), accounting for more than than half of the total gene expression variation even in experiments involving harsh treatments such as heat stress or reciprocal transplantation (Dixon et al., 2015, Dixon et al., 2018). Interindividual variation in gene expression also contributes to variation in stress responses (e.g., Granados-Cifuentes et al., 2013 and Barshis et al., 2013).

We have added lines to the introduction to more clearly establish our objective and rationale.

Lines 70-74 : “Previous studies have identified interindividual differences in gene expression under benign conditions that may contribute to observed differences in stress responses (Granados-Cifuentes et al., 2013 and Barshis et al., 2013. The genes the expression of which is associated with resistance to bacterial challenge and calcification over the subsequent year enhance our understanding of the molecular determinants of disease resistance and growth in coral.”

And then, if you were looking for correlations between gene expression levels and resistance/susceptibility to bacterial infection, why did you not collect tissue during the experiment, for example after the 50% tissue loss had been reached, or from replicated fragments made before the start of the experiment ?

Firstly and most importantly, we were looking for correlations between *predictive* gene expression and resistance/susceptibility. Therefore, sampling a replicate fragment at the start of the experiment is appropriate.

We would like to offer some insight as to why, in this case, the reviewer's suggestions would not yield convincing results even **if** our objective was to find differentially regulated genes after exposure (which it was not).

Suggestion for a response experiment #1) "Collecting tissue when 50% mortality had been reached"...

(1) Our previous work has shown that by the time a lesion has sufficiently progressed (or even appeared) on a fragment of this size, the entire fragment is dying. The tissue is beginning to fall off the skeleton. The RNA is incredibly degraded. The gene expression profile we would likely be seeing is "death", not "response to disease." Ongoing work with other stressors (e.g., heat stress) provide growing evidence that it's important to sample for a gene expression response before symptoms are apparent, as by the time symptoms are apparent you have likely missed the "stress-specific" response and are seeing more general declines in health (e.g., cell death). We have added lines to the discussion to discuss this ongoing research.

Lines 271-283: "One objective of the TagSeq experiment was to associate baseline differences in gene expression with subsequent exposure outcome. We did not intend to measure differences in gene expression *responses* between individuals and, in this case, we would be unable to do so since we had to wait until the mortality was well-apparent, at which point the affected coral fragments were not suitable for meaningful gene expression analysis. Still, identifying differences in baseline gene expression responses between susceptible and resistant individuals is just as critical to understanding molecular mechanisms underlying disease tolerance. Future studies attempting to fill that knowledge gap will need to carefully consider when to sample for expression. For example, recent work investigating responses to thermal stress in corals and symbiotic anemones has shown that symbiosis breaks down well before bleaching becomes apparent (Radecker et al., 2021; Cleves et al., 2020).

Our study to identify predictive gene expression associations with disease outcome Gene expression analysis revealed subtle differences in pre-exposure transcriptomic states between corals that subsequently demonstrated resistance or susceptibility to *Vibrio* challenge."

(2) The other challenge with designing a response expression study that samples gene expression when a particular sample reaches a phenotypic threshold point (e.g., 50% mortality, some level of bleaching...) when you are interested in differences between individuals is that, by definition of your study, different individuals will reach those values at different times. Differences in gene expression would be confounded by sampling time point and resistance metric.

Suggestion for a response experiment #2) “Sampling from replicated fragments”... Now, this would certainly work for a response experiment indeed, that is exactly what we did in two previously published experiments that set out to measure differences in expression responses (Wright et al., 2017 and Wright et al., 2019). Please understand that that was not the objective here.

This is a real missed opportunity to produce valuable gene expression data that would allow us to discover what genes may be involved in making certain genets resistant. I highly encourage you to pursue this question and try again by setting up an experiment that will get you some results.

Our results are the outcome of an experiment designed to answer our stated objective: to associate inter-individual variation in baseline gene expression with variation in their resistance to bacterial challenge and long-term growth. The experiments proposed by the reviewer are suitable for an experimental objective to compare gene expression responses to bacterial challenge and we encourage others to pursue this important work.

Here, one differentially expressed gene between resistant and susceptible genets cannot be yielding significant GO enrichment ! I do not understand how you get to those results ?

The methods section explains how we performed the GO enrichment using GO-MWU, which applies a Mann-Whitney U test to the whole ranked list of genes. The test determines whether genes belonging to a specific GO term are significantly clustered near the top or bottom of the list. Sidestepping the need to impose an arbitrary “significance cutoff” is one of the major advantages of the GO_MWU method, which also makes it completely valid in situations like ours, where very few genes are individually identified as genome-wide significant.

The result that resistant genets do not seem to show a decrease in growth rate after bacterial exposure is interesting, but it would have been nice to clearly see those results in the paper.

These results are stated in the results section (lines 190-195), and we also had a supplementary figure about that. Now, following the reviewer’s request to show the results more clearly, we have moved that figure into the main manuscript. We hope this change satisfies the reviewer’s critique.

Is there really no growth difference between the control and exposed resistant fragments after a year?

Perhaps surprisingly, that is indeed what we have observed. These results are stated in the results section (lines 190-195) and in what was previously Supplemental Figure 5 and now Figure 3.

Finally, some of your main conclusions are unfortunately not supported by your results. When you state : "Predictive (pre-exposure) gene expression highlights subtle differences between resistant and susceptible genets" and "Furthermore, gene expression signatures associated with resistance and long-term growth help inform strategic assessment of coral health parameters.", it is all based on that one single differentially expressed gene. This lack of gene detection is not because of a technical error, but because you are most likely not looking in what/when you need to. Because the pathogen resistance side of the paper is half of the story, I cannot recommend your manuscript for publication in the present form. I know it is disappointing, but I would suggest completing this work with data that compare gene expression levels between resistant and susceptible genets under bacterial assault, or re-write the paper to bring the predictive genes and long-term growth rate correlation side in the fore front. But, I would agree it does not constitute a very compelling story in itself without additional work.

Best wishes

We hope our previous responses and some clarifying paragraphs to the manuscript can convince the reviewer that we did not set out to “compare gene expression levels between resistant and susceptible genets under bacterial assault” and that our actual objective, to associate baseline differences in expression profiles with the individual-specific resistance to bacterial challenge and long-term growth, has been met with our results, even if we did not achieve a scientifically thrilling result of strongly associated predictive genes. We believe that this result is just as important for understanding the molecular underpinnings of coral health as would be a result describing differences in responses. Also, since we are using rank-based GO analysis, our functional insights are not based on a single significant gene, but on the overall pattern of (relatively subtle) gene expression shifts among all genes.

Appendix B

Associate Editor Comments to Author:

The editors recommend acceptance, following the comments of the reviewer, but we would also like you to respond to the remaining comments from that reviewer to more fully address the potential limitations of the work (and how these could/should be overcome in future works). This need not be an extensive re-write, but adding a few sentences to take these points into consideration would be welcome.

We again thank the editorial team for their continued coordination. We have added several sentences to the discussion that we believe address these important concerns and help guide future studies.

Reviewer comments to Author:

Reviewer: 2

Comments to the Author(s)

Thank you for your efforts in responding to my questions and concerns. I can concede that you set this study to look for constitutive gene expression that would predict the outcome of pathogen assaults. However, as you very pertinently pointed it out, gene expression variation between corals of the same population and species is often great, therefore in order to answer your main question of interest, the sample size would have to be much larger for a chance to detect a signal (if it exists). I was just surprised this was your experimental design choice to go after that kind of question. Your paper is well written and the analyses are sound. I just remain unconvinced that your sample size was ever going to let you detect constitutive gene expression differences predictive of disease resistance among genets living in the same general environmental conditions. To give you an example, recent inflammasome markers on different immune cell populations can predict the clinical outcome of COVID patients, but these markers were evaluated on 66 people. But I also understand that balancing budget, sample size and question of interest can be tricky. Best wishes.

We thank the reviewer for their thoughtful critique. We have added the following sentences to the discussion to address these concerns and guide future studies.

Line 281–287: “Furthermore, future studies should note that our ability to detect predictive gene expression associated with a subsequent bacterial challenge outcome was limited by our sample size ($n = 26$ genets in the gene expression analysis). Collection limits for threatened species and experimental constraints, such as isolated aquaria needed to prevent coral-to-coral disease transmission, restrict the number of genets that can be included in similar experiments. We recommend careful consideration of the sample size necessary to detect the signature of interest given large variation in constitutive gene expression among individuals.”